# Physical Mapping of the *Anopheles* (*Nyssorhynchus*) *darlingi* Genomic Scaffolds

**DOI:** 10.3390/insects12020164

**Published:** 2021-02-15

**Authors:** Míriam Silva Rafael, Leticia Cegatti Bridi, Igor V. Sharakhov, Osvaldo Marinotti, Maria V. Sharakhova, Vladimir Timoshevskiy, Giselle Moura Guimarães-Marques, Valéria Silva Santos, Carlos Gustavo Nunes da Silva, Spartaco Astolfi-Filho, Wanderli Pedro Tadei

**Affiliations:** 1Coordenação de Sociedade Ambiente e Saúde, Laboratório de Vetores de Malária e Dengue, Instituto Nacional de Pesquisas da Amazônia, Av. André Araújo, 2936, Manaus, AM 69060-001, Brazil; wptadei@gmail.com; 2Programa de Pós-Graduação em Genética, Conservação e Biologia Evolutiv, Instituto Nacional de Pesquisas da Amazônia, Manaus, AM 69060-001, Brazil; lcbridi@gmail.com (L.C.B.); moura.giselle@gmail.com (G.M.G.-M.); santos.val.silva@gmail.com (V.S.S.); 3Department of Entomology and Fralin Life Science Institute, Virginia Polytechnic Institute and State University, Blacksburg, VA 24061, USA; igor@vt.edu (I.V.S.); msharakh@vt.edu (M.V.S.); v.a.timoshevsky@gmail.com (V.T.); 4Laboratory of Evolutionary Genomics of Insects, the Federal Research Center Institute of Cytology and Genetics, Siberian Branch of the Russian Academy of Sciences, 630090 Novosibirsk, Russia; 5Department of Genetics and Cell Biology, Tomsk State University, 634050 Tomsk, Russia; 6MTEKPrime, Aliso Viejo, CA 92656, USA; omarinotti@gmail.com; 7Department of Biology, University of Kentucky, Lexington, KY 40506, USA; 8Programa de Pós-Graduação em Biotecnologia, Universidade Federal do Amazonas, Av. Rodrigo Otávio, 6.200. Coroado l, Manaus, AM 69080-900, Brazil; cgmanaus@gmail.com; 9Laboratorio de Tecnologias de DNA, Divisão de Biotecnologia, Centro de Apoio Multidisciplinar, Universi dade Federal do Amazonas, Av. Rodrigo Otávio, 6.200. Coroado l, Manaus, AM 69080-900, Brazil; spartaco.biotec@gmail.com

**Keywords:** in situ hybridization, genomics and cytogenetics, synteny, polytene chromosome

## Abstract

**Simple Summary:**

*Anopheles darlingi* mosquitoes are the main vectors of malaria in the Brazilian Amazon. To assign genomic DNA sequences to chromosomes of this species, we performed fluorescence in situ hybridization of DNA probes with salivary glands polytene chromosomes. We compared the physical locations of the *An. darlingi* probes with homologous sequences in other *Anopheles* species, namely *Anopheles albimanus* and *Anopheles gambiae*. The results demonstrated that substantial genome rearrangements occurred throughout the evolutionary history of these mosquitoes. The physical mapping data can be useful for improving the structural accuracy of the *An. darlingi* genome assembly and for understanding the chromosomal evolution of these mosquitoes.

**Abstract:**

The genome assembly of *Anopheles darlingi* consists of 2221 scaffolds (N50 = 115,072 bp) and has a size spanning 136.94 Mbp. This assembly represents one of the smallest genomes among *Anopheles* species. *Anopheles darlingi* genomic DNA fragments of ~37 Kb were cloned, end-sequenced, and used as probes for fluorescence in situ hybridization (FISH) with salivary gland polytene chromosomes. In total, we mapped nine DNA probes to scaffolds and autosomal arms. Comparative analysis of the *An. darlingi* scaffolds with homologous sequences of the *Anopheles albimanus* and *Anopheles gambiae* genomes identified chromosomal rearrangements among these species. Our results confirmed that physical mapping is a useful tool for anchoring genome assemblies to mosquito chromosomes.

## 1. Introduction

The *Anopheles* genus includes vector species of great importance to public health, such as those transmitting malaria parasites [1,2,3]. This genus contains seven subgenera, of which two are the focus of the present study: *Nyssorhynchus* Blanchard, which includes 39 species, among them the neotropical malaria vectors *An. darlingi* and *Anopheles albimanus,* and *Cellia* Theobald with 224 species, including *An. gambiae* [4].

*Anopheles darlingi,* the subject of this study, is the major contributor to malarial transmission in the Amazonian region of South America [5,6,7,8]. The distribution of the species reaches from Southern Mexico to Northern Argentina and from East of the Andean Mountains to the coast of the Atlantic Ocean [1,9,10]. The importance of *An. darlingi* as a malaria vector spurred effort for genome sequencing, assembling and annotation [11].

While the size of the *An. darlingi* haploid genome was determined by cytometric analysis to be ∼201 Mb (2C = 0.41 pg), sequencing and assembling resulted in an *An. darlingi* fragmented, draft genome that spans only 173.9 Mb. The difference between the cytometrically determined genome size and the sum of all the contigs and scaffolds is most likely the result of unassembled centromeres, telomeres and other portions of the genome that are rich in repetitive DNA sequences. In fact, 18.66% of the reads were not included in the final assembly [11]. Presently, the only draft *An. darlingi* genome accessible in VectorBase is composed of 2221 scaffolds. While long-read sequencing technologies are currently available, efforts for chromosome-level reference genome assembly are still lacking for this medically important mosquito species.

Chromosomal physical maps and genomic sequencing both contribute to accurate genome assembling [12]. The FISH method is a useful tool for the development of chromosome-anchored assemblies and correcting scaffold arrangements [13]. By using the FISH technique, several DNA sequences have been mapped in the *An. darlingi* chromosomes. Positions of gene-specific sequences for rDNA [14], heat shock protein (Hsp) 70 [15], actin [16], myosin [17], glutathione S transferase (GST) [8], and Gram-negative bacteria binding protein (GNBP) [18] have been placed onto the *An. darlingi* photomap [19]. In this study, we expanded the number of probes hybridized to *An. darlingi* chromosomes, mapping nine additional DNA sequences, originating from fosmid clones. The physical map we are building will support an improved, more complete and more ordered genome assembly for *An. darlingi*.

## 2. Material and Methods

### 2.1. Mosquitoes

Gravid *An. darlingi* females were captured at Bairro do Puraquequara (3°5.19′5″ S and 59°8.92′62″ W), Manaus, Amazonas State, Brazil. They were captured from 6:30 to 9:00 p.m. when resting on stable walls or by human landing catches (HLC), by trained technicians using personal protective equipment. The collection of specimens was authorized by the Chico Mendes Institute for Biodiversity Conservation–ICMBio and Biodiversity Information and Authorization System–SISBIO, Brazil, through permanent license number 32941 (21 May 2012), for the collection of zoological material, issued to Dr. Míriam Silva Rafael.

Morphological identification of specimens was carried out according to taxonomic keys [20,21]. Captured *An. darlingi* gravid females were confined individually in plastic cups for egg laying. Offspring were fed with powdered fish food (Tetramin fish food (Tetramin) was purchased from local shop (Tetramin Tropical Flakes-Spectrum Brands, Inc).) and reared to the fourth instar larvae.

### 2.2. Chromosome Preparation

Salivary glands of fourth instar *An. darlingi* larvae were dissected in Carnoy’s solution (100% ethanol: glacial-acetic-acid, 3:1) and then fixed with Fixative I (Carnoy’s solution: water, 1:5) for 3 to 5 min, Fixative II (Carnoy’s solution: water, 1:1) for 3 min and Fixative III (95% lactic acid: water, 1:1) for 5 to 8 min [15,22,23]. The samples were then placed on a slide and crushed with a coverslip. After removing the coverslip, the slides were flash frozen in liquid nitrogen and immediately placed in cold 50% ethanol. After that, preparations were dehydrated in an ethanol series (70%, 90%, and 100%) for 3 min each, air-dried and stored at 20 °C.

### 2.3. Probes Preparation and Fluorescence in Situ Hybridization (FISH)

A fosmid library with inserts containing 30–40 kb of *An. darlingi* DNA was prepared according to the Copy Control fosmid Library Production (EPICENTRE) Kit protocol. The fosmid library was constructed as part of the *An. darlingi* genome project [11], using genomic DNA extracted from mosquitoes captured in Coari, Amazonas State, Brazil.

Fosmid clones were randomly selected, and DNA was isolated by standard laboratory methods [24] and then labeled with 5-Amino-propargyl-2’-deoxyuridine 5’-triphosphate coupled to Cy^3^ fluorescent dye (Cy^3^-AP^3^-dUTP), or 5-Amino-propargyl-2’-deoxyuridine 5’-triphosphate coupled to Cy^5^ fluorescent dye (Cy^5^-AP3-dUTP), (GE Healthcare U.K. Ltd., Chalfont St Giles, UK) by nick translation following the manufacturer’s instruction (Thermo Fisher Scientific, Inc. Waltham, MA, USA).

Slides were fixed in ethanol battery and 4% paraformaldehyde in 1 × phosphate-buffered saline (PBS) following denaturation of the target and probe DNA at 90 °C. The labeled probes were hybridized to the chromosomes at 37 °C overnight and then slides were washed with 0.2× saline sodium citrate (SSC) buffer. We detected fluorescent signals using YOYO-1 *Invitrogen* 895247 (Invitrogen Corporation, Carlsbad, CA, USA).

### 2.4. Physical Mapping and Probe Location Analysis

Preparations of carefully selected photos of different chromosomes of salivary glands from *An. darlingi* were considered representative of the banding pattern on 10 slides. All slides were photographed at 100× objective and 10×/25 objective (Carl Zeiss MicroImaging, Inc., Thornwood, NY, USA) with an Axiocam MRC charge-coupled device camera (Zeiss) using the Axiovision program (version 3.1). The probe localization was determined within a subdivision, using a standard cytogenetic map for *An. darlingi* [19,25]. The microphotographs were edited with Adobe Photoshop CS4 (Adobe Systems Incorporated, San Jose, CA, USA) and, also an Olympus CX41 phase-contrast microscope (Olympus America Inc., Melville, NY, USA). We also performed the comparison of synteny of sequences mapped in *An. darlingi* by in silico against *An. gambiae* and *An. albimanus*, since their genomes have already been sequenced and their scaffolds analyzed.

### 2.5. DNA Sequencing and Bioinformatics Analyses

The 3′ and 5′ end sequences (400–700 bp) of the inserts contained in each fosmid clone were determined using Sanger sequencing (ABI 3730XL sequencer) and vector-based primers (Copy Control fosmid Library Production EPICENTRE). Vector sequences were removed, and the remaining sequences were analyzed by the BLASTN tool against the *An. darlingi* genome assembly AdarC3 (www.vectorbase.org/blast) available at VectorBase [26]. Syntenic regions in the genomes of *An. albimanus* and *An. gambiae* were identified by the Comparative Genomics tool available at VectorBase. Genes contained in probes were assumed to be those located between each of the end sequences when they were in the same scaffold of the assembly AdarC3. When the end sequences of a probe were assigned to different scaffold, we took a conservative approach, considering only those genes located within 5 kb of the BLAST hits.

## 3. Results

### 3.1. Probes and Corresponding Sequences in the Genome Assembly

In this work, we used fosmid clones as DNA probes for physical chromosome mapping. The fosmid DNA library was constructed previously as part of the *An. darlingi* genome project [11]. The main advantage of using fosmid clones for FISH is that the large size of labeled probes allowed us to obtain strong and specific fluorescent signals on polytene chromosomes. In most of the *An. darlingi* polytene chromosome preparations assayed by FISH, strong fluorescent signals were observed for the probes used in this study. The genomic scaffolds associated with each fosmid probe were determined by BLASTN using the insert’s end DNA sequences as query against the *An. darlingi* genome assembly AdarC3 (Appendix A). In general BLAST analyses resulted in a single hit within the *An. darlingi* genome. The probe Pb2r retrieved three hits with similar e-values, scores, and percentage identity. The Pb2r sequence resulting in multiple hits corresponds to the ADAC001061, ADAC003002 and ADAC001064, three ~6 kb long genes, annotated as vitellogenins and sharing extremely high nucleotide identity. Pb2r and Pb20b had one of their end sequences retrieving no hits, indicative of regions that were not assembled during the AdarC3 genome assembling efforts. The end sequences of Pb17r and Pb18b resulted in BLAST hits located in different scaffolds, (Pb17r-DGSJ01A09C.g00 = scaffold_1409 and DGSJ01A09C.b02 = scaffold_1350) (Pb18b- DGSJ02B03C.b00 = scaffold_683 and DGSJ02B03C.g00 = scaffold_1062).

### 3.2. Chromosome X

Of the nine DNA probes, none of them mapped to the chromosome X of *An. darlingi*.

### 3.3. Chromosomes 2 and 3

Some of the probes produced multiple signals in Chromosomes 2 and 3. Pb2r mapped in regions 8A, 10A and 10E, of 2R (Figure 1). Pb5r-Cy^3^, mapped in 15B inside 2nd inversion close to 2Rc inversion in the pericentromeric 2R, and in the 31A region of 3R, in a paracentric 3Ra inversion, which belongs to a complex of inversions (3Rb, 3Rb, 3Rc) (Figure 2). The Pb7b-Cy^5^ mapped in the band 6A of 2R in *An. darlingi* (Figure 3), a location that is homologous with 2R chromosome tip in *An. gambiae.* Pb17r-Cy^3^ hybridized in situ on 2R (9A) and 2L (25C), and Pb18b-Cy^5^ mapped in 2L (region 16B) (Figure 4). Pb19r-Cy^3^ and Pb20b-Cy^5^ were probed in the same slide. Pb19r-Cy^3^ hybridized to 25C of 2L, near the telomeric region, while Pb20b-Cy^5^ hybridized in 2L (25B and 21D) and 3R (27B) (Figure 5). Pb22b-Cy5 mapped in the 43C region of 3L, into the inversion 3La. This inversion belongs to a complex set of 3La, 3Lb, 3Lc inversions in the chromosome arm 3L (Figure 6). Pb23r-Cy^3^ mapped in 2L band 22C, in the 2La inversion, which belongs to a complex of 2La, 2Lb inversions in the paracentromeric 2L (Figure 7).

### 3.4. Comparative Mapping

The majority of the mapped probes showed conserved chromosomal location in the species *An. darlingi*, *An. albimanus*, and *An. gambiae* (*An. darlingi=An. albimanus=An. gambiae*, X=X=X, 2R=2R=2R, 2L=2L=3L, 2L=2L=3R, and 3L=3L=2L (Table 1). The probes with correspondences 2R=2R=2R were Pb2r-Cy3 (scaffolds_732, 1942 and 1482), Pb5r-Cy3 (scaffold_112:91297-130051), and Pb7b-Cy5 (scaffold_281:57476-88809). The *An. darlingi* probes with correspondence 2L=3L=2L were Pb19r-Cy3 (scaffold_17:336283-371424) and Pb23r-Cy3 (scaffold_17:435950-474554). The Pb17r-Cy3 (scaffolds 1409 and 1350) and Pb18b-Cy^5^ (scaffolds 683 and 1062) probes’ equivalence was 2L=2L=3R. Finally, Pb22b-Cy^5^ (scaffold_17:435950-474554) results showed equivalence 3L=3L=2L.

## 4. Discussion

*Anopheles darlingi,* the subject of this study, is the major malaria vector in the Amazonian region of South America [5,6,7,8]. Its importance as a vector spurred studies of *An. darlingi* biology, cytogenetics, behavior, physiology, biochemistry, genetics, and insecticide resistance [7,11,14,15,19,27,28,29,30,31,32,33,34,35,36]. Here we expanded the knowledge of the physical chromosomal map of this species and compared the results in silico with homologous sequences in two other anopheline species, *An. albimanus* and *An. gambiae*. *Anopheles albimanus,* belonging to the same subgenus as *An. darlingi*, *Nyssorhynchus*, is distributed in the Neotropical region stretching from the southern United States to northern Peru and the Caribbean islands. This species is the major contributor to malaria transmission in the coastal areas of these regions [13]. The evolutionary divergence between *An. darlingi* and *An. albimanus* was estimated at ~40 million years [37]. *Anopheles gambiae* belongs to another subgenus, Celia, and is the major vector of *Plasmodium falciparum* in Africa [38]. The evolutionary relationship and divergence time of *An. darlingi* in comparison with *An. gambiae* was estimated at ~100 million years [39].

The first records related to the chromosomal maps and inversions polymorphisms in *An. darlingi* were obtained in the 1950s [40,41,42]. Later, a more extensive study described nine independent inversions and a complex arrangement [43]. They analyzed two Brazilian populations, one from the Amazon region and another from the south of the country, finding that the population from the north is highly polymorphic when compared to the population from the south. In populations of *An. darlingi* from Manaus, about 90% of the analyzed polytene nuclei showed one or more heterozygous inversions [43]. Furthermore, an inversion on chromosome 2 (2Rd), one on chromosome 3 (3Rc), and one on X chromosome (Xb) from populations of *An. darlingi* captured during the rainy season around the BR-174, Manaus-Boa highway Vista, State of Amazonas, were described [44]. In addition, breakpoints of paracentric inversions and additional inversions (3Lc), totaling 18 inversions in the chromosome arms of *An. darlingi* with one on chromosome X, seven on chromosome 2, and 11 on chromosome 3 were previously described [25]. Because of the highly polymorphic *An. darlingi* chromosomes, cytogenetic photomaps are essential tools for standardized conclusions. A photomap of *An. darlingi* chromosomes was developed for a population of Guajará-Mirim, State of Rondônia, Brazil [19]. The *An. darlingi* cytogenetic photomap included sections and subsections of chromosomes and their description. These features of the photomap allowed us to place genomic sequences to specific chromosomal positions.

Five of the nine probes used in this study hybridized to a single location in the *An. darlingi* chromosomes, producing an unequivocal position assignment. Among them is Pb18b, for which end sequences aligned with two different scaffolds. This result supports the conclusion that scaffolds 683 and 1062 are closely located.

The remaining four probes hybridized to two or three distinct locations. These results may indicate that these probes contain repetitive DNA sequences and/or genes recently duplicated with high similarity among them. Pb2r, for example contains sequences of the Vitellogenin genes. Multiple Vitellogenin genes have been identified in several mosquito species. Duplication, concerted evolution, and purifying selection have been identified as major evolutionary forces driving Vitellogenin genes’ evolution and conserved sequences [45]. Pb17r had end sequences aligned with two different scaffolds, supporting the conclusion that scaffolds 1350 and 1409 are closely located. 17r and 18b end sequences blast hits are positioned close to the ends of the scaffolds, further supporting the conclusion of their adjacent locations within the genome.

Three probes—Pb19r, Pb22b and Pb23r— were assigned to the scaffold 17, however, Pb19r and Pb23r hybridized to 2L, while Pb 22b hybridized to 3L. Scaffold 17 is a long scaffold, and these results suggest it is the result of misassembled sequences. Pb19r and Pb23r are located at least 300,000 kb apart from Pb22b, which is located between coordinates 39978 and 73280. Alternatively, a translocation event between the populations of Coari, which originated the assembled genome, and the population from Manaus, used in this study, may be the cause of the result. Polymorphisms of inversions, fusions, and translocations in chromosomal arms are among the determining factors of the local adaptation of mosquitoes under heterogeneous conditions [46,47,48].

A previous synteny evaluation between *An. darlingi* and *An. gambiae* identified 1027 synteny clusters, comprising 6312 syntenic genes or ∼60% of all *An. darlingi* protein-coding genes [11]. However, the synteny clusters were not assigned to chromosome arms, and it was recommended that mapping of genes or clones on chromosomes, together with the described synteny clusters, would support a more complete and precise assembly of the *An. darlingi* genome.

Our results demonstrate a complex picture in which probes Pb2r, Pb5r and Pb7b indicate a conservation of the 2R arm among *An. darlingi, An. albimanus* and *An. gambiae.* Probes hybridizing to *An. darlingi* 2L, however, have syntenic regions scattered among 2L, 3R, and X in *An. albimanus* and *An. gambiae*.

## 5. Conclusions

We mapped nine DNA probes to *An. darlingi* polytene chromosomes and compared them by in silico analysis with *An. albimanus* and *An. gambiae* genomes. Our results highlighted the necessity of additional efforts to improve and achieve a more complete, chromosome-level *An. darlingi* genome assembly. Malaria remains a major healthcare risk in South America, and a chromosome-level reference genome of *An. darlingi* will help in developing successful vector management approaches and the understanding of vector evolution using comparative genomics.

## Figures and Tables

**Figure 1 insects-12-00164-f001:**
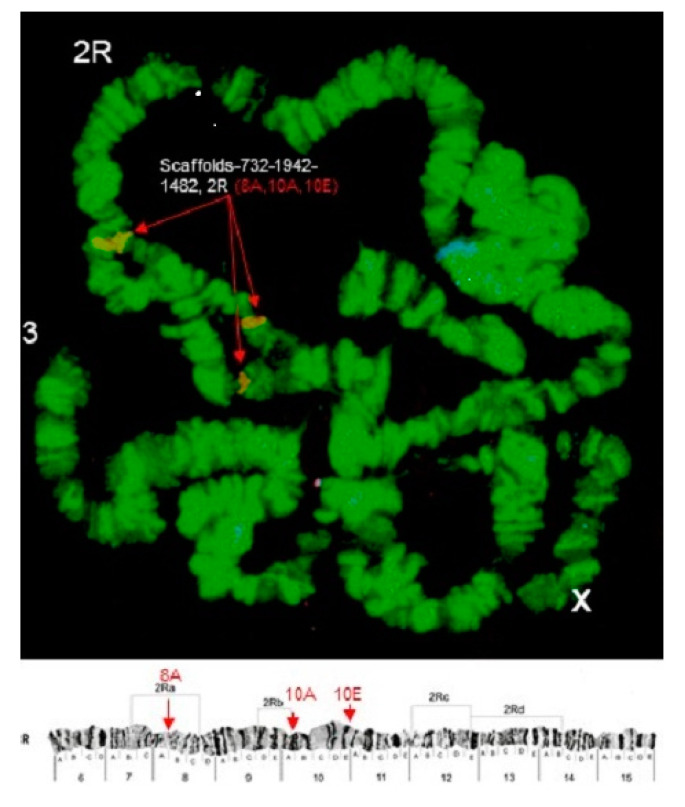
Pb2r-Cy3 probe, mapped in 2R (8A,10A, and 10E) indicated by arrows. 2Ra, 2Rb, 2Rc and 2Rd show inversions in 2R [19]. Chromosomal bands with positive hybridization are identified with text in red, and the corresponding scaffolds in *An. darlingi* genome assembly AdarC3 are indicated in white text.

**Figure 2 insects-12-00164-f002:**
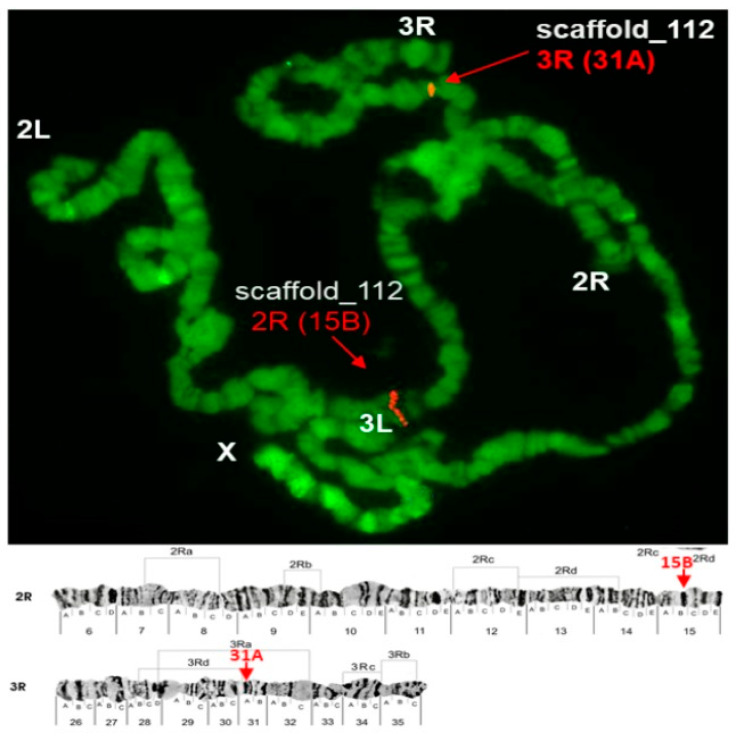
Pb5r-Cy3 mapped in 2R (15B) and 3R (31A), as shown by arrows. Chromosomal bands with positive hybridization are identified with text in red, and the corresponding scaffolds in *An. darlingi* genome assembly AdarC3 are indicated in white text.

**Figure 3 insects-12-00164-f003:**
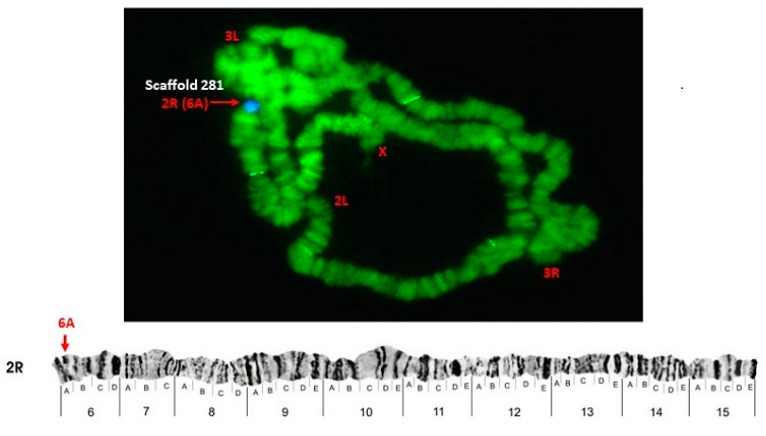
Pb7b-Cy^5^ probe, mapped in the 6A band in 2R of *A. darlingi*, as shown by arrows. Chromosomal bands with positive hybridization are identified with text in red, and the corresponding scaffolds in *An. darlingi* genome assembly AdarC3 are indicated in white text.

**Figure 4 insects-12-00164-f004:**
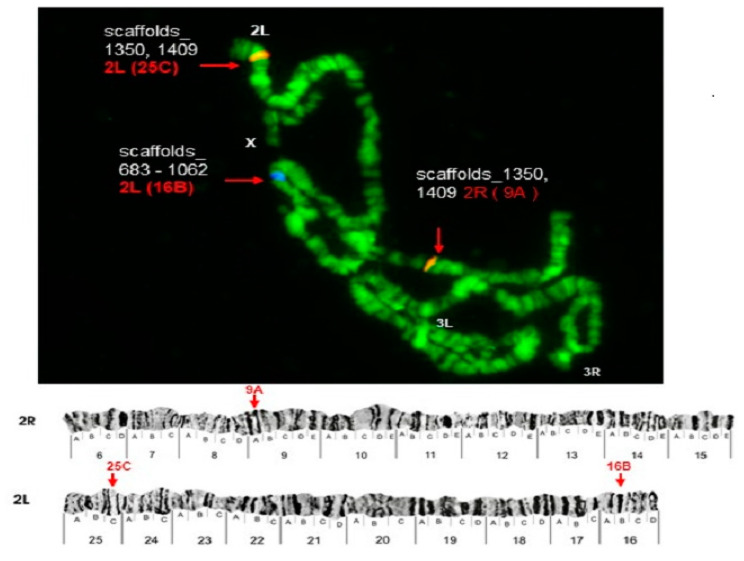
Pb17r-Cy^3^ hybridized in situ on 2R (9A) and 2L (25C), and Pb18b-Cy^5^ probe, mapped in 2L (region 16B). Chromosomal bands with positive hybridization are identified with text in red, and the corresponding scaffolds in *An. darlingi* genome assembly AdarC3 are indicated in white text.

**Figure 5 insects-12-00164-f005:**
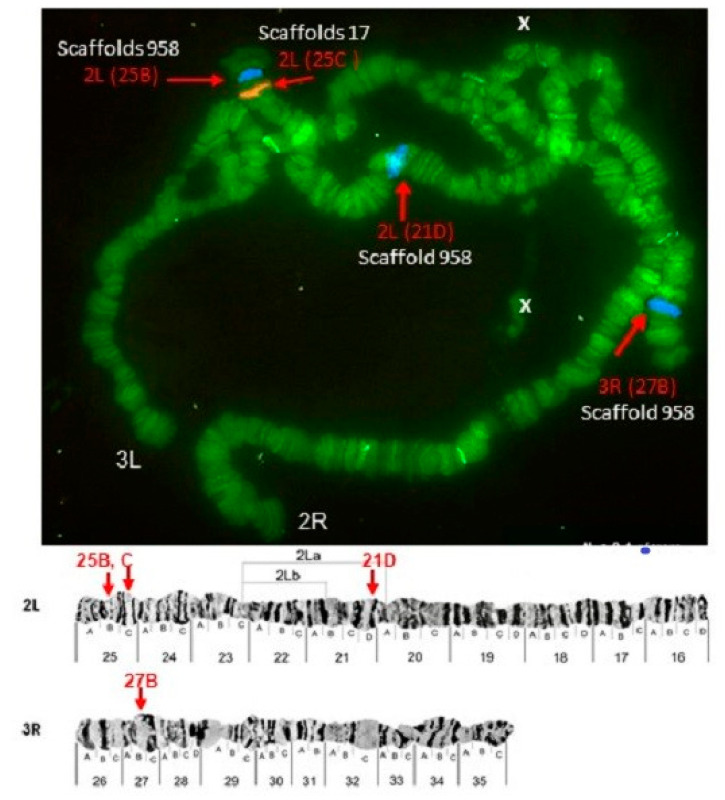
Pb19r-Cy^3^ probe, mapped in 2L (25C) and Pb20b-Cy^5^ probe mapped in 2L (21D and 25B) and 3R (27B). Chromosomal bands with positive hybridization are identified with text in red, and the corresponding scaffolds in *An. darlingi* genome assembly AdarC3 are indicated in white text.

**Figure 6 insects-12-00164-f006:**
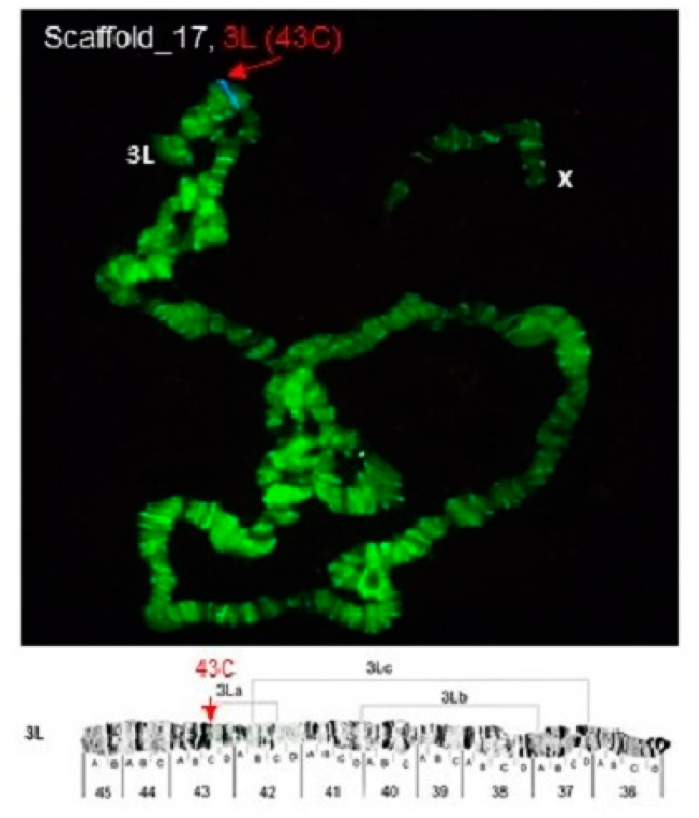
Pb22b-Cy^5^ probe, mapped in 3L (43C), a region involving the chromosomal inversion 3La in *An. darlingi,* as shown by arrows [19]. Chromosomal bands with positive hybridization are identified with text in red, and the corresponding scaffolds in *An. darlingi* genome assembly AdarC3 are indicated in white text.

**Figure 7 insects-12-00164-f007:**
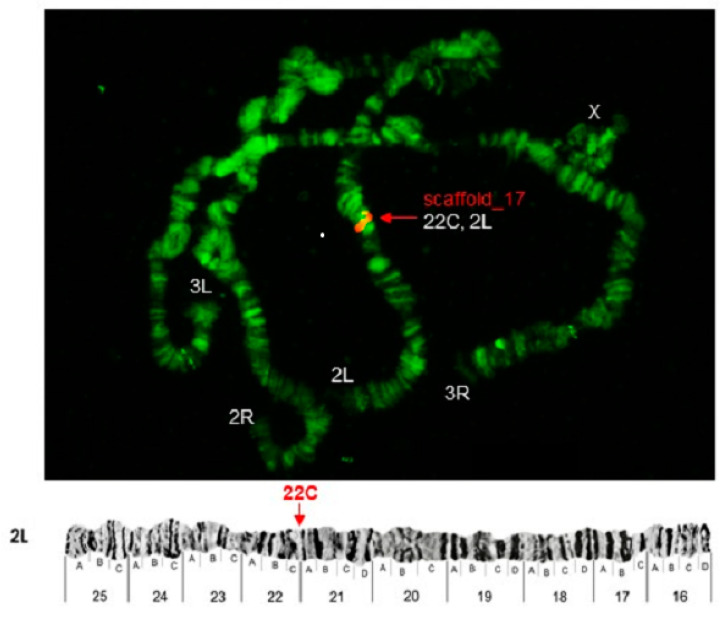
Pb23r-Cy^3^ probe mapped in 2L (22C). Chromosomal bands with positive hybridization are identified with text in red, and the corresponding scaffolds in *An. darlingi* genome assembly AdarC3 are indicated in white text.

**Table 1 insects-12-00164-t001:** In situ localization of probes in *An. darlingi* chromosomes and in silico identification of equivalent sequences in *An. albimanus* and *An. gambiae* chromosomes. Details of BLAST hits locations within *An. darlingi* scaffolds and probes FASTA files are available in Appendix A.

Probe.	BLAST Hits	*In Situ* Chromosome Mapping *Anopheles darlingi*	*In Silico chromosome* Mapping *Anopheles albimanus*	*In Silico* Chromosome Mapping *Anopheles gambiae*
Pb2r	scaffold_732	2R (8A, 10A, 10E)	2R (10A)	2R (18B)
	scaffold_1942			
	scaffold_1482			
Pb5r	scaffold_112	2R (15 B), 3R (31A)	2R (12C)	2R (12C)
Pb7b	scaffold_281	2R (6A)	2R (10B)	2R (13B)
Pb17r	scaffold_1409	2L (25C), 2R(9A)	2L (24A)	3R (32D)
	scaffold_1350			
Pb18b	scaffold_683	2L (16B)	2L (17A)	3R (29A)
	scaffold_1062			
Pb19r	scaffold_17	2L (25C)	3L (45A)	2L (21D)
Pb20b	scaffold_958	2L (21D,25B), 3R (27B)	X (1A)	X (5B)
Pb22b	scaffold_17	3L (43 C)	3L (45A)	2L (28C)
Pb23r	scaffold_17	2L (22C)	3L (45A)	2L (23D)

## Data Availability

Data is contained within the article or Appendix A.

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
