# Peer review of "Physical Mapping of the Anopheles (Nyssorhynchus) darlingi Genomic Scaffolds"

_insects, 2021, doi:10.3390/insects12020164_

Round 1
Reviewer 1 Report
Rafael and colleagues report the mapping of 9 fosmid clones to salivary gland polytene chromosomes in the malaria vector Anopheles darlingi. A polytene chromosome map for this species was published in 2010 and a genome assembly reported in 2013. Gene synteny in Anopheles species has been reported in several studies. The figures are very clear and well presented. I see no need to do any additional experiments.
The text however is hard to follow and would benefit from a much sharper focus and proof reading by a native English speaker. I’ll highlight a few sentences that need to be rewritten but the whole text needs careful review.
As I understand it, the aim is to physically map 9 fosmid clones to better link the genome assembly to the polytene chromosome map. This had been done by others with single genes. It was not clear to me how much we have gained by mapping these 9 fosmid clones.
Major points:
Results
- A brief sentence or two at the beginning to explain to the reader the approach taken. Why did you go to the trouble of making a fosmid genome library? They must provide an advantage
- Why were the 9 fosmid clones selected for mapping?
- Was each entire fosmid clone sequenced completely or just end-sequenced? I have not worked with fosmids but my understanding is that they are about 40kb. In the Table S1 the query length is less than 1kb. So the fosmid clones were only end sequenced? Also the column heading should be length.
- Is the entire 40kb clone used as the probe?
- Why do some fosmid clones map to more than one scaffold? Is the entire 40 kb fosmid in more than one scaffold or just a small region? Table 1 suggest the overlap is small, less than 1kb. Are there regions of the genome duplicated?
- Fig 3. Why does the fosmid clone map to 3 locations? Is this a large duplicated region or just a multi gene family?
- Table S1. The length of nt similarity and % identity to the other Anopheles species should be listed.
- Line 171. “Pb2r-Cy3 (scaffolds_732-1942-1482:2322:20557), Pb5r-Cy3 (scaffold_112:91297-130051)”. I guess this is saying Pb5r-Cy3 has been sequenced completely and matches about a 39 kb region of scaffold 112. But I have no idea what is being said about Pb2r-Cy3.
- Fig 9. The chromosomes look well aligned. I don’t see any evidence of an inversion as indicated in the drawing.
- Much of the discussion is on the genes in the 9 fosmid clones. The genes are not mentioned in the results. Supp tables 2 and 3 are not cited in the text. Tables 2 and 3 could be combined. When saying why these 9 fosmid clones were selected at the beginning of the results, the genes in each probe could be noted. Or they could be noted when describing the results of the in situ hybridization with each probe.
Discussion
This is overly long for what is a short report on the in situ mapping of 9 clones to polytene chromosomes. The authors should confine the discussion to the significance of their results in this study. How does the mapping of these 9 clones improve our understanding of the genome of this species? It was not clear to me from reading the discussion. For example, the final paragraph begins “We mapped nine DNA probes by FISH to An. Darlingi” but then rather than place the importance of the mapping of these 9 probes the paragraph goes on to summarize the results of other Anopheles genome projects.
Line 357. What is the point of this paragraph? Could be deleted.
Line 388. “In the present work, the BBS2 (Pb17r-Cy3 probe, scaffolds:1409_1350) of An. darlingi hybridized in situ on its chromosome 2R (band 9A) and 2L (band 25C), whose scaffolds are located next to each other.” What? Why did the probe hybridize to two bands? I have no idea what “Pb17r-Cy3 probe, scaffolds:1409_1350” means.
Line 398. “Paracentric inversions or translocations in chromosome arms are the only large-scale rearranges that occur in the evolution of Anopheles genome [2]. This event could explain the Pb18b-Cy5 (scaffold 683_1062) probe of An. darlingi, which was mapped in silico in the 16B region near the centromeric region of 2L and in the 2L arm (17A region) in An. albimanus.” What? 2L 16B in An. Darlingi and 2L 17A in An. Albimanus. That doesn’t seem that big a difference to justify discussion of a paracentric inversion without mapping other nearby genes.
Line 407: “In An. darlingi Pb20b-Cy5 (scaffold_958) was physically mapped in arms 2L (21D, 25B), and 3R (27B), which is located on both X (11A region) chromosomes of An. albimanus and X (5B) of An. gambiae.” Why is Pb20b-Cy5 mapping to 3 locations in An. darlingi? The authors conclude that this is evidence of a translocation or transposition from the X chromosome to 2L and 3R. This would seem unlikely. Is it possible that the probe is cross-reacting with a related gene that is on 2L and also 3R? How long is the sequence homology with the other Anopheles species? This observation needs a more critical discussion.
Line 421. In this study one probe containing an OBP gene mapped to 2L, whereas in a previous study a different OBP gene was mapped to the X chromosome. I think that is what this paragraph is saying. If so, so what? There are many OBP genes in the genome. Is it a surprise that one is on the X chromosome and one on 2L? I don’t understand the point that is being made in this paragraph.
Author Response
To
Editor of Insects
Re: Revised manuscript titled “Physical Mapping of the Anopheles (Nyssorhynchus) darlingi Genomic Scaffolds”.
Pleaase find our revised version of the manuscript. According to the reviewer’s comments the manuscript is now shorter and more focused on the results of the hybridizations. Furthermore, the new text highlights the necessity of additional efforts to improve and achieve a more complete, chromosome level An. darlingi genome assembly. Our results will support a new chromosome-level reference genome of An. darlingi and will help in developing successful vector management approaches and the understanding of vector evolution using comparative genomics.

Reviewer 2 Report
Review of insects-1058945-v1
Scientific premise is not clearly presented. Please revise manuscript to clearly present what is known, what is unknown and how this study improved the previously published genome assembly.
I found the inclusion of An. albimanus in this study puzzling and the justification is not clearly laid out. The story seems to be more clearly presented if only gambiae and darlingi are included. If the two species are used because they are only a few of mosquito genomes with chromosome-level assembly, it should be clearly stated as such in the introduction with citation https://pubmed.ncbi.nlm.nih.gov/32883756/.
Some instruction and large chunk of discussion may indicate that the contribution of this work could be understanding of chromosome inversions and translocation among Anopheles species. However, I found it difficult to understand based on the results presented here. The 9 probes seem inadequate to present any paracentric chromosome inversions. Figure like Neafsey et al. 2015 Fig 2A could be useful in presenting the results and help readers to understand the topic of inversion/translocation.
Major revisions
Lines 36-37: “Our results confirmed that physical mapping is a useful tool 35 for developing chromosome-anchored genome assemblies.” – The numbers shown in the abstract does not show the improvement of genome assembly toward chromosome-anchoring. To indicate the benefits of approach more clearly, the paper should present the number without FISH (number of scaffolds and/or N50) and improved assembly stats after FISH. The numbers in Marinotti et al. 2013 https://www.ncbi.nlm.nih.gov/pmc/articles/PMC3753621/ should be mentioned to quantify the improvements, discrepancy between the two studies clearly presented in the abstract.
Lines 42-43. Provide example citations after sentence.
Lines 60-61: “Despite its importance in malarial transmission, no previous reports of successful 60 colonization of An. darlingi …” This is in conflict with the following previous reports: https://pubmed.ncbi.nlm.nih.gov/26024853/
https://pubmed.ncbi.nlm.nih.gov/31340377/
Please revise the sentence to reconcile with the fact that there is some success in colonizing An. darlingi previously.
Lines 61-65: The more important numbers to provide here is the genome size, the number of scaffolds, and N50 from citation 14 that needs to be compared with the results of this study.
Figure 1: font size too small.
Figure 2: small line drawing should have chromosomal arm label for clarity. Check with Cornel et al. https://www.ncbi.nlm.nih.gov/pmc/articles/PMC3753621/ for the updated chromosome map based on population of Manaus, Brazil. Perhaps use different colors for fonts and known gene names? The study used 9 mapped probes but there are more than 9 things on the chromosome map and it is not clear which are actual probes mapped.
Figure 3-4, 6: font size too small. Unusually large border. Weblink font much larger than the other text.
Figure 5, 7-8: font size too small.
(Only figure 9 has adequate font size)
Supplemental Table: The table should contain information of gambiae ortholog’s chromosome locations for each ADAP gene or each probe and their chromosome location based on FISH to clearly show how synteny is preserved
Lines 79-87: Topic of chromosome inversion seems unrelated to the main results presented here. And evidence of improving our understanding of chromosome structure is not clearly presented.
Minor revisions
Lines 43-47: Break up the sentence for clarity.
Line 47: Change ‘An. albimanus’ to ‘Anopheles albimanus’ – A sentence should not start with genus abbreviation.
Lines 78-79: “Distribution of An. darlingi orthologous genes were encoded along An. gambiae chromosome 2R.” – It is not clear what this means or if this is important in the context of gene organization. I think the more important message to be included in this paragraph is that gene content in a chromosome appear to be preserved between Anopheles darlingi and An. gambiae despite 100Myrs of separation. In other words, orthologs of genes on a scaffold of darlingi appears to be on the same chromosome of Anopheles gambiae.
Lines 84-85: “However, An. darlingi from the Brazilian Amazon showed a high degree of polymorphic inversions” – The most up-to-date chromosome inversion map should be cited here https://pubmed.ncbi.nlm.nih.gov/27223867/
Author Response

(The authors gave the same response as above.)

Round 2
Reviewer 1 Report
The manuscript has been significantly improved.
I have only a few minor comments for the authors to address.
- As I mentioned in my first review, I think a couple of sentences could be added for the general reader on the advantages of fosmid clones as cytological markers -with citations (eg note stability as single copy). It could also be mentioned here that a fosmid DNA library was made previously made as part of the genome project.
- The authors have clarified that only the ends of the fosmid clones were sequenced. However, I don’t believe it is correct to say “flanking sequence”. That would mean the sequence that is flanking or outside of the fosmid clone. The clones were sequenced at each end with the sequences about 40 kb apart.
- The 9 clones used as probes were apparently selected at random. This is good to know but is not stated in the text. Please note in methods.
- Line 361 “An. darlingi were made [40-42]”. Made by? Some extra text is needed here
- Line 368. polytene not polythene
- The discussion is much more focused on the results of this study. Two of the probes, 17r and 18b, had end sequences that blast hit to different scaffolds. This is discussed for 18b on line 381. The explanation is possible but only if the blast hits are to the ends of the scaffold. If the hit is more than 40 kb (length fosmid) from the end of the scaffold then I think genome misassembly is a better explanation. That 17r blast hit two scaffolds should also be discussed in this section.
- Line 407. I think the last paragraph of the discussion is really two sentences. The long sentences should be broken after discussing the conservation of 2R among the three species.
The evidence from just 9 probes highlights the need for a better genome assembly. eg PacBio HiFi of DNA from a single individual. So, I agree with the conclusions.
Author Response
Response to Reviewer 1 (round 2)
Comments and Suggestions for Authors
The manuscript has been significantly improved.
I have only a few minor comments for the authors to address.
Point 1. As I mentioned in my first review, I think a couple of sentences could be added for the general reader on the advantages of fosmid clones as cytological markers -with citations (eg note stability as single copy). It could also be mentioned here that a fosmid DNA library was made previously made as part of the genome project.
Response 1: We added explanatory sentences in the beginning of the paragraph 3.1. Probes and corresponding sequences in the genome assembly.
Point 2. The authors have clarified that only the ends of the fosmid clones were sequenced. However, I don’t believe it is correct to say “flanking sequence”. That would mean the sequence that is flanking or outside of the fosmid clone. The clones were sequenced at each end with the sequences about 40 kb apart.
Response 2: We replaced “flanking sequence” with “end sequence” throughout the text.
Point 3. The 9 clones used as probes were apparently selected at random. This is good to know but is not stated in the text. Please note in methods.
Response 3: We noted this information in methods.
Point 4. Line 361 “An. darlingi were made [40-42]”. Made by? Some extra text is needed here
Response 4: Corrected.
Point 5. Line 368. polytene not polythene
Response 5: Corrected.
Point 6. The discussion is much more focused on the results of this study. Two of the probes, 17r and 18b, had end sequences that blast hit to different scaffolds. This is discussed for 18b on line 381. The explanation is possible but only if the blast hits are to the ends of the scaffold. If the hit is more than 40 kb (length fosmid) from the end of the scaffold then I think genome misassembly is a better explanation. That 17r blast hit two scaffolds should also be discussed in this section.
Response 6: A sentence recognizing that 17r hits two scaffolds and the possible explanation was inserted in the discussion. The reviewer is correct that in the cases of 17r and 18b the hits are close to the ends of the scaffolds, further supporting the conclusion of adjacent locations within the genome. This information was also included in the discussion.
Point 7. Line 407. I think the last paragraph of the discussion is really two sentences. The long sentences should be broken after discussing the conservation of 2R among the three species.
Response 7: The sentence is split in two sentences.
The evidence from just 9 probes highlights the need for a better genome assembly. eg PacBio HiFi of DNA from a single individual. So, I agree with the conclusions.

Reviewer 2 Report
Line 81-82: GPS coordinate degree, minute and second symbols appears to be messed up. Could be due to character encoding error but needs to be fixed before publication.
Figure 9. Fonts on Figure 9 seems too small. Editorial decision needed to confirm that this complies with journal standard.
Lines 366-367. The timeline doesn’t add up. A photomap published in 2010 and the new subsections were described and added in 1972? Perhaps wrong citation?
Author Response
Response to Reviewer 2 Comments (round 2)
Comments and Suggestions for Authors
Point 1. Line 81-82: GPS coordinate degree, minute and second symbols appears to be messed up. Could be due to character encoding error but needs to be fixed before publication.
Response Point 1. They look fine in our file.
Point 2. Figure 9. Fonts on Figure 9 seems too small. Editorial decision needed to confirm that this complies with journal standard.
Response Point 2. The figure was replaced with a table.
Point 3. Lines 366-367. The timeline doesn’t add up. A photomap published in 2010 and the new subsections were described and added in 1972? Perhaps wrong citation?
Response Point 3. This section was rewritten.
